# Mechanism of *cis*-Nerolidol-Induced Bladder Carcinoma Cell Death

**DOI:** 10.3390/cancers15030981

**Published:** 2023-02-03

**Authors:** Mateo Glumac, Vedrana Čikeš Čulić, Ivana Marinović-Terzić, Mila Radan

**Affiliations:** 1Department of Immunology and Medical Genetics, School of Medicine, University of Split, 21000 Split, Croatia; 2Department of Medicinal Chemistry and Biochemistry, School of Medicine, University of Split, 21000 Split, Croatia; 3Department of Biochemistry, Faculty of Chemistry and Technology, University of Split, 21000 Split, Croatia

**Keywords:** nerolidol, sesquiterpenes, cell death pathways, calcium signaling, ER stress

## Abstract

**Simple Summary:**

The aim of this study was to determine the antitumor activity of nerolidol, a naturally occurring sesquiterpene alcohol, in two bladder carcinoma cell lines and to elucidate the possible cell death pathway. Bladder carcinoma is one of the most common malignancies worldwide, with a high recurrence rate and low survival. In our study, we observed multiple cellular mechanisms of nerolidol with two distinct modes of cell death. Its specific induction of cell death, which does not depend on apoptotic factors, is a potential therapeutic approach to overcome the inherent resistance of tumors to apoptosis.

**Abstract:**

Nerolidol is a naturally occurring sesquiterpene alcohol with multiple properties, including antioxidant, antibacterial, and antiparasitic activities. A few studies investigating the antitumor properties of nerolidol have shown positive results in both cell culture and mouse models. In this study, we investigated the antitumor mechanism of *cis*-nerolidol in bladder carcinoma cell lines. The results of our experiments on two bladder carcinoma cell lines revealed that nerolidol inhibited cell proliferation and induced two distinct cell death pathways. We confirmed that *cis*-nerolidol induces DNA damage and ER stress. A mechanistic study identified a common cAMP, Ca^2+^, and MAPK axis involved in signal propagation and amplification, leading to ER stress. Inhibition of any part of this signaling cascade prevented both cell death pathways. The two cell death mechanisms can be distinguished by the involvement of caspases. The early occurring cell death pathway is characterized by membrane blebbing and cell swelling followed by membrane rupture, which can be prevented by the inhibition of caspase activation. In the late cell death pathway, which was found to be caspase-independent, cytoplasmic vacuolization and changes in cell shape were observed. *cis*-Nerolidol shows promising antitumor activity through an unorthodox mechanism of action that could help target resistant forms of malignancies, such as bladder cancer.

## 1. Introduction

Nerolidol (3,7,11-trimethyl-1,6,10-dodecatrien-3-ol), also known as peruviol, is an aliphatic sesquiterpene alcohol with an asymmetric center at the C-3 position and a double bond at the 6-position, which makes it exist in four stereoisomeric forms. It occurs naturally in the essential oils of various plants with a floral odor, usually as a mixture of all four isomers, with (+)-*trans*-nerolidol as the predominant form [1,2]. Nerolidol is declared safe for human consumption by the FDA [2]. It is used to enhance the flavor and aroma of different food products, but it is also used as a fragrance in perfumes, cosmetics, and other household products [3]. Nerolidol exhibits antioxidant [4,5,6,7], antinociceptive [8], antimicrobial [9,10,11,12,13], antiparasitic [14,15,16,17], and antitumor [10,18,19,20,21] activities. Nerolidol has low water solubility (1.532 mg/L at 25 °C) [2], leading to its incorporation into cell membrane bilayers, possibly by orienting along phospholipids, making them more fluid. Making membranes more fluid increases transmembrane drug penetration, but an excessively high increase in fluidity induces membrane breakdown [22,23,24]. Regarding nerolidol isoforms, (+/−)-*cis*-nerolidol has been proven to be more cytotoxic than (+/−)-*trans*-nerolidol in multiple tumor cell lines [19]. The exact mechanism of the antitumor activity of nerolidol has not yet been elucidated; however, its effects on multiple cellular mechanisms have been observed. A more pronounced effect is the ability to alter the mitochondrial membrane potential (MMP) by both *cis*- and *trans*- isoforms [19,22,23]. *cis*-Nerolidol induces endoplasmic reticulum (ER) stress [19] by inducing a misfolded protein response via the up-regulation of ERN1 and EIF2AK3. It also reduces the transcription of pro-apoptotic proteins BAK1, BAX, CAPN1, CASP8, CASP9, PAPR1, and TP53. The transcription of proteins involved in the cell cycle, CCND1, CCNE1, CDK1, and CDK2, showed that nerolidol could alter the cell cycle, which was confirmed by flow cytometry showing the accumulation of cells in the G1 phase [20]. *trans*-Nerolidol increases the expression of the adhesion proteins E-cadherin and β-catenin, reducing the tumor’s ability to metastasize [20]. It showed no effect on the activation of caspases, but in combination with tumor necrosis factor α (TNFα), it significantly increased caspase activation compared to TNFα alone, potentiating apoptosis. It also suppresses TNF-α-induced NF-κB phosphorylation [20]. Few animal studies have investigated nerolidol activity. A study conducted on F344 rats showed that nerolidol had a protective effect against the development of adenomatous polyps and adenocarcinomas [25]. Other studies have reported a reduction in oxidative stress in neuronal cells [26] and hepatoprotective capabilities [4]. 

Bladder carcinoma (BC) is one of the most prevalent cancers worldwide. Most BC cases are associated with environmental factors, with tobacco smoke being the leading cause [27]. Men have a 4-fold higher incidence, with 80% of cases diagnosed after the age of 65 [27]. BC originates from the urothelial cells that form the inner layer of the urinary bladder. It can present as non-muscle invasive bladder carcinoma (NMIBC), muscle-invasive bladder carcinoma (MIBC), or as a metastatic form of the disease [28]. BC is infamous for its high recurrence rate and disease progression, despite successful initial treatment [29]. NMIBC is the most common form of the disease, accounting for up to 75% of all cases [30,31]. The treatment of NMIBC involves trans-ureteral resection and intravesical implementation of post-operative therapy in the form of Bacillus Calmette–Guérin (BCG) or classical chemotherapeutics, such as mitomycin C, cisplatin derivates, 5-fluorouracil, and others [28,29]. The advantage of intravesical therapy is the sparing of the rest of the body from toxicity caused by the applied agent [28]. The rate of disease recurrence for NMIBC following surgical intervention and post-operative therapy is very high (30–80%), and the disease progression of 10–45% emphasizes the need to target BC at this stage [30,31]. MIBC is treated with peri-operative platinum-based chemotherapy, followed by radical cystectomy [28]. Not only does radical cystectomy reduce the quality of life for the patient, but the progression of the disease to MIBC also leads to a high chance of the development of metastasis (50–70% of MIBC patients develop metastasis), leading to a low survival rate (5-year survival rate is less than 5%) [32]. The last major breakthrough in the prevention of BC recurrence after partial cystectomy was the instillation of BCG in the bladder, which started in 1977 and is still the gold standard [28]. Few advancements in available therapeutics that can successfully combat BC have made it one of the least survivable malignancies [33]. In the last decade, our molecular understanding of BC has greatly improved treatment outcomes [28]. The introduction of checkpoint inhibitors has revolutionized metastatic BC therapy [28]; however, their effectiveness is not universal, and they can only be employed in a specific subgroup of patients [29]. The lack of predictive markers that could identify which patients would respond better to checkpoint inhibitors led to the preservation of classical chemotherapy as the first-line treatment [29]. The development of more successful chemotherapies that can be employed earlier in treatment could help prevent BC recurrence and progression. 

As the number of BC cases increases owing to environmental factors and the aging population [27], there is a need for better treatment options. As BC is usually diagnosed at an early stage [27], more successful early intervention could reduce recurrence rates and prevent disease progression. This study aimed to investigate the antitumor activity of *cis*-nerolidol in human BC cell lines for its potential use as a therapeutic agent for BC. Our goal was to identify the signaling pathways involved in nerolidol-induced cell death and determine whether it could be exploited for combating BC. 

## 2. Materials and Methods

### 2.1. Chemicals

*cis*-Nerolidol (from here written as nerolidol) was obtained from Sigma-Aldrich (St. Louis, MO, USA) (72180). Small molecular inhibitors: xestospongin C (XSC; 64950), dandrolene-Na (DRN; 14326), SB203508 (13067), H-89 (10010556), KH7 (13243), and L755507 (18629) were obtained from Cayman Chemicals. Z-Vad (FMK001) was obtained from R&D systems. All chemicals were dissolved in dimethyl sulfoxide (DMSO; Sigma-Aldrich; D4540). MTT (M5655) and trypan blue (T8154) were obtained from Sigma-Aldrich.

### 2.2. Cell Culture

T24 and TCCSUP cells were obtained from the American Type Culture Collection (ATCC) and cultured in Dulbecco’s modified Eagle’s medium (D6429, Sigma-Aldrich) with 10% fetal bovine serum (F0804, Sigma-Aldrich) and 100 U of penicillin/streptomycin (P0781, Sigma-Aldrich) in a humidified incubator at 37 °C in a 5% CO_2_ atmosphere. Trypsin-EDTA (T4049; Sigma-Aldrich) was used for cell passage and division. For nerolidol treatment, nerolidol was mixed with the complete growth medium to obtain the indicated concentrations. For experiments involving the use of inhibitors, cells were treated with inhibitors dissolved in complete growth medium at the indicated concentrations for 1 h before nerolidol administration: H-89 (50 nM), z-Vad (20 or 40 μM), SB203580 (0.5, or 4 μM), DRN (10 μM), XSC (380 nM), KH7 (3 and 10 μM), and L755507 (50 pM or 600 nM). All chemicals were mixed with complete growth medium and treatments were applied by replacing the existing growth medium to ensure homogenous administration. For each treatment, an appropriate control was used, which was an equivalent amount of vehicle (usually DMSO). Microscopy was performed using an Olympus CHX41 microscope (Olympus Corporation, Tokyo, Japan) and cell photographs were taken using an SC50 camera (Olympus Corporation). 

### 2.3. Cell Counting and Trypan Blue Staining

The cells were seeded in 24- or 12-well plates and grown overnight. Treatments were performed in triplicate with different concentrations of nerolidol (25, 50, 60, 75, and 100 mg/L) indicated for each experiment. Incubation times were 0.5, 2, 4, 24, 48, or 72 h of treatment, as indicated for each experiment. For experiments involving inhibitors, cells were pre-treated with inhibitors for 1 h before nerolidol administration. Cell counting was performed using an Olympus CHX41 microscope (Olympus Corporation) using a Neubauer chamber. Cell viability was assessed by staining cells with trypan blue stain after obtaining a single cell suspension. Cells were counted as described before. Cells stained blue were considered not viable. Data are presented as a share of trypan blue-stained cells in a total cell count (100%) for each measurement. 

### 2.4. Annexin-V/PI Staining

TCCSUP cells were treated with nerolidol (50 mg/L) for 48 h and analyzed using the Annexin V-FITC Apoptosis Detection Kit I (BD Biosciences, San Jose, CA, USA; 556547) following the manufacturer’s instructions. Briefly, after treatment with nerolidol (50 mg/L), the cells were trypsinized, washed with PBS, and resuspended in 100 µL of binding buffer containing 5 µL annexin V-FITC and/or 5 µL propidium iodide (PI). The cells were incubated for 15 min at room temperature in the dark and analyzed using flow cytometry (BD Accuri C6, BD Biosciences). The percentages of annexin V- and PI-positive cells were analyzed using FlowLogic Software (Inivai Technologies, Mentone VIC, Australia; https://flowlogic.software/flowlogic/, accessed on 15 March 2021) and are presented as mean values ± SD. 

### 2.5. Hydrogen Peroxide Levels in Nerolidol-Treated Cells (H_2_O_2_-Glo)

Intracellular ROS levels were measured using the ROS-Glo H_2_O_2_ assay (Promega Corp., Madison, WI, USA, G8820) following the manufacturer’s instructions. Briefly, TCCSUP cells were treated with nerolidol (50 mg/L) for 24 h. During the final 6 h of incubation, 20 μL of the H_2_O_2_ substrate solution was added to the cells. At the end of the incubation period, ROS-Glo detection solution was added, incubated for 20 min, and luminescence was measured using a Synergy/HTX multi-mode microplate reader (BioTek, Winooski, VT, USA).

### 2.6. Measurement of Cytosolic Ca^2+^

Continuous calcium dynamics after nerolidol exposure were measured as previously described [34]. Briefly, TCCSUP cells were pre-treated with DRN, XPS, or both for 1 h. They were then trypsinized and stained with 3μM Fluo-3-acetoxymethyl ester (Cayman Chemicals, Ann Arbor, MI, USA, 14960) in suspension for 30 min. Cells were treated with nerolidol (50 and 100 mg/L) in suspension. The fluorescence was analyzed using an Accuri C6 flow cytometer (530/30 filter). 

### 2.7. Determination of ATP

Changes in the intracellular ATP levels were measured using a luciferin/luciferase system. To determine the ATP levels, TCCSUP cells were seeded in 96-well plates at a density of 1 × 10^4^ cells/well. The cells were exposed to nerolidol (2, 8, 25, 125, and 500 µM) for 20 and 60 min, after which they were lysed and treated with a luciferin/luciferase mixture containing ATP. Luminescence was measured using a Synergy/HTX Multimode Microplate Reader (Agilent, Santa Clara, CA, USA).

### 2.8. Measurement of cAMP

Changes in the intracellular cAMP levels were determined by the cAMP-Glo^TM^ Assay (V1501, Promega) following manufacturer’s instructions. Briefly, TCCSUP cells were seeded in 96-well plates at a density of 1 × 10^4^ cells/well and treated with 50 mg/L of nerolidol for 60 min. Cells were washed with PBS and lysed in cAMP-Glo™ Lysis Buffer for 15 min. Then, cAMP-Glo™ Detection Solution supplemented with Protein Kinase A was added and incubated for 20 min. The Kinase-Glo^®^ Reagent was added and the mixture was incubated for 10 min. Luminescence was measured using a Synergy/HTX Multimode Microplate Reader (Agilent, Santa Clara, CA, USA).

### 2.9. SDS-PAGE and Western Blot Analysis

T24 and TCCSUP cells were treated with 50 and 75 mg/L nerolidol and incubated for 2, 24, and 48 h. After incubation, cells were lysed using RIPA lysis buffer supplemented with protease (Complete tablets Easy pack 04693116001, Roche (Basel, Switzerland)) and phosphatase (PhosSTOP EASYpack 04906837011, Roche) inhibitors, benzonase (E1014, Sigma), and 5 mM MgCl_2_. Lysates were then centrifuged at 16,000× *g* at 4 °C. To prepare samples for SDS-PAGE, the lysates were combined with 6× Laemmli buffer and heated to 95 °C for 5 min. Electrophoresis was conducted on polyacrylamide gels after which transfer onto nitrocellulose membranes was performed using an electrophoresis method. Membranes were blocked with TBS containing 5% BSA and 0.1% NaN_3_. Immunoblotting was performed using various antibodies. CASP9 (# 9508), cleaved-CASP3 (# 9661), phospho-p44/42 MAP kinase (# 9101), p38 MAPK (# 9212), PARP (# 9542S), pMLKL (# 91689S), p42/44 MAP kinase (# 9102 L), and yH2Ax (# 9718S) were obtained from Cell Signaling Technology; E2F1 (sc-251), cyclin A (sc-751), GRP78 (sc-166490), HSP90 (sc-69703), ATF4 (sc-390063), and GADD153 (sc-7351) were obtained from Santa Cruz Biotechnology; LC3 (PM036) was obtained from MBL International Corporation; and α-tubulin (TUBA) (049K4767), and β-actin (ACTN) (A5316) were obtained from Sigma-Aldrich. Cyclin D1 (M3642) was obtained from Agilent Dako. Detection was performed using ChemiDoc (Universal Hood II; Bio-Rad Laboratories, Inc., Hercules, CA, USA). 

### 2.10. Alkaline Comet Assay

The cells were trypsinized and resuspended in PBS. The resuspended cells were mixed with 2% low-melting-point (LMP) agarose (cooled to <40 °C) and layered on top of an agarose-covered slide (1%). After 10 min polymerization time on ice, cell lysis was performed in the gel with the lysis buffer (2.5 M NaCl, 100 mM EDTA, 10 mM Tris-HCl, Triton 1%, and DMSO 10%) for 1 h at 4 °C. After lysis, the slides were washed with cold phosphate-buffered saline (PBS). Before electrophoresis, the slides were incubated in 300 mM NaOH with 1 mM EDTA for 10 min, and then subjected to electrophoresis in the same buffer (300 A, 20 min, ~30 V). The slides were neutralized with 50 mM Tris-HCl (pH 7.5) for 10 min and stained with ethidium bromide for 5 min. After staining, the comets were visualized using an Olympus CHX41 microscope (Olympus Corporation).

### 2.11. Statistical Analysis

Statistical analysis was performed using GraphPad Prism (version 9) built-in functions. Experiments containing three or more groups with one variable were analyzed using a one-way ANOVA. A post hoc analysis was performed using Tukey’s multiple comparison tests between any two groups. Two-way ANOVA was performed for multiple variable experiments. Post hoc analysis was performed using Dunnett’s or Sidak’s multiple comparison tests. Only results with *p* < 0.05 were considered significant in any performed analysis. Symbols for different significance levels were assigned as follows: ns, *p* > 0.05; * *p* < 0.05; ** *p* < 0.001; *** *p* < 0.0001; and **** *p* < 0.00001. All data for measured variables are expressed as mean ± SD. The sample size used for statistical tests was *n* = 3 (if not indicated differently). Half maximal inhibitory concentration (IC_50_) for the reduction in cell count was calculated from normalized data using the [Inhibitor] vs. normalized response—variable slope model. Calculated IC_50_ values that were higher than the largest dose used were considered not reliable and were reported only as higher than the largest tested concentration.

## 3. Results

### 3.1. Nerolidol Induces Two Different Mechanisms of Cell Death with Distinct Cell Morphologies

To test how nerolidol influences cell count, two bladder carcinoma cell lines (T24 and TCCSUP) were treated with increasing concentrations of nerolidol (25, 50, 75, and 100 mg/L). The cells were counted at multiple time points (4, 24, 48, and 72 h post-treatment), and the results are presented in Figure 1A. Nerolidol inhibited proliferation and reduced the cell count, evident from 24 h onward, in a concentration-dependent manner. Both the T24 and TCCSUP cell lines responded similarly to the treatment, with the TCCSUP cell line being more resistant to nerolidol. The data obtained in this experiment was used to calculate the IC_50_ for the reduction in cell count (Appendix A). The results showed that lower concentrations of nerolidol were necessary to reduce the cell numbers by 50% as the duration of treatment increased. Trypan blue staining was used to distinguish the dead cells from the live cells. Staining revealed two distinct periods of cell death, as shown in Figure 1B, corresponding to 4 and 48 h after treatment. The highest dose of nerolidol (100 mg/L) caused staining in almost 90% of T24 cells and 70% of TCCSUP cells 4 h after treatment; however, the total cell count was preserved (approximately 90% for both cell lines). Lower doses (25 mg/L and 50 mg/L) did not cause a significant increase in staining at the 4 h time point. At 24 h, trypan blue staining was significantly reduced (two-way ANOVA, *p* < 0.0001), followed by high cell staining at 48 h, in a concentration-dependent manner. Examples of trypan blue staining are shown in Appendix A. To further validate the early cell death period, T24 cells were exposed to increasing concentrations of nerolidol (50, 75, and 100 mg/L) and stained with trypan blue at 0.5 and 2 h after treatment. Although no significant loss in cell count was observed (two-way ANOVA, *p* = 0.9341) (Appendix A), the number of trypan blue-stained cells increased in a concentration- and time-dependent manner, confirming the loss of cell viability shortly after nerolidol treatment (Appendix A).

Cell counting revealed that, in addition to inducing cell death, nerolidol inhibited cell proliferation. This was evident from the slower increase in cell counts at treatment doses that did not induce a significant level of cell death (25 and 50 mg/L). To test whether nerolidol disturbed cell cycle progression, the cell cycle markers E2F1, cyclin A, and cyclin D1 were immunoblotted in nerolidol-treated cells (T24 and TCCSUP). All tested markers were strongly reduced, particularly at higher nerolidol concentrations (Figure 1C). 

Cell morphology also changed dynamically following nerolidol treatment (Appendix A) in both tested cell lines. Some cells exhibited cell membrane blebbing, followed by swelling and rupture of the cell membrane, leading to early cell death. Other cells that did not exhibit this phenotype developed distinct cytoplasmic vacuolation at later time points, and the cell shape changed from round to more square-like and then elongated (fibroblast-like). Cells that died during the late cell death period did not exhibit membrane blebbing or creation of apoptotic bodies, and the cell membrane was preserved. Cytoplasmic vacuolation was confirmed by immunoblotting for LC3, which showed increased LC3 expression and processing at 24 h in both cell lines (Figure 1D). A schematic representation (with examples) of the described changes in cell morphology during the early and late cell death periods is presented in Figure 1E,F, respectively.

### 3.2. Prolonged Nerolidol Exposure Caused Increasing Levels of DNA Damage and ER Stress

The T24 and TCCSUP cells were treated with 50 and 75 mg/L nerolidol for 2, 24, and 48 h. The 2 and 48 h time points were selected to represent the periods of cell death while the 24 h time point was selected as an intermediary point between the two cell death periods. The nerolidol concentrations used for experiments were chosen according to trypan blue staining as 50 mg/L mostly induced the late cell death event while 75 mg/L efficiently induced the early cell death event but left enough cells to observe the induction of the late cell death event. Cell lysates were analyzed by WB using phosphorylated H2Ax (γH2Ax) as a marker of DNA damage. The results showed an increase in γH2Ax at all time points and in both cell lines compared to control cells (Figure 2A). γH2Ax increased with the treatment duration, suggesting the accumulation of DNA damage. To confirm this, a comet assay was performed on T24 cells treated with 60 mg/L nerolidol for 2 and 24 h. The results showed the same pattern: DNA breaks were present and increased with treatment duration (Figure 2B). T24 and TCCSUP cells treated with 50 and 75 mg/L nerolidol for 2, 24, and 48 h exhibited cytoplasmic vacuolation, commonly associated with ER stress. To test whether nerolidol induced ER stress during the early and late cell death, the ER stress markers GRP78, HSP90, and ATF4 were immunoblotted (Figure 2C). We observed a mild increase in GRP78 protein levels, particularly at the 48 h time point, in T24 cells. HSP90 increased in TCCSUP cells at both time points in the 50 mg/L treatment, whereas it only increased at the 2 h time point in T24 cells in the 75 mg/L treatment. The ER stress-linked transcription factor ATF4 increased at both tested time points, and the effect was dose-dependent. In T24 cells, it increased for 50 mg/L treatment at both time points, but not at 75 mg/L, whereas in TCCSUP cells, it increased for both concentrations.

### 3.3. Nerolidol Cytotoxicity Originates from Ca^2+^ Depletion in the ER through Ryanodine Receptors (RYR)

Nerolidol exerts both acute and prolonged effects on cells. Prolonged nerolidol treatment led to cytoplasmic vacuolation and cell shape transformation, even at concentrations that did not induce significant cell death. This effect may be the result of high cytoplasmic Ca^2+^ levels [35]. To determine whether cytoplasmic calcium (Ca^2+^) is an early event in nerolidol cytotoxicity, cytoplasmic Ca^2+^ levels were measured in TCCSUP cells after nerolidol administration (Figure 3A). Nerolidol treatment induced a large increase in cytoplasmic calcium levels, similar to thapsigargin, an inhibitor of the ER calcium ATPase pump. To elucidate the source of Ca^2+^, cells were treated with XSC, an inositol-1,4,5-trisphosphate receptor (IP3R) inhibitor, DRN, a ryanodine receptor (RYR) inhibitor, or their combination, for 1 h before measurement. The experiment was performed in Ca^2+^-free PBS to avoid the possibility of Ca^2+^ influx through Ca^2+^ channels located on the cell membrane. The results showed that the inhibition of IP3R did not prevent an increase in cytoplasmic Ca^2+^ levels after treatment. RYR inhibition completely prevented the increase in cytoplasmic Ca^2+^ at 50 mg/L nerolidol while an additional 100 mg/L nerolidol (150 mg/L in total) only marginally increased the Ca^2+^ levels. The results for the combination of both inhibitors did not differ from those for the RYR inhibitor. Flow cytometry results were confirmed by trypan blue staining and cell counting. T24 cells were pre-treated with DRN for 1 h before nerolidol administration. Nerolidol (100 mg/L) was administered to induce a high rate of early nerolidol-induced cell death. Cells were stained and counted after 2 h of treatment. DRN almost completely inhibited the loss of cell viability (Figure 3B). Cells pre-treated with DRN did not exhibit cell blebbing or swelling compared to non-inhibited, nerolidol-treated cells (Figure 3C). To test whether DRN inhibited late nerolidol-induced cell death, T24 cells were pre-treated with DRN for 1 h and then treated with 60 mg/L nerolidol for 48 h to induce a high proportion of late-induced cell death. Cells were stained with trypan blue and counted. DRN almost completely inhibited the loss of cell viability (Figure 3D) and prevented cytoplasmic vacuolation and cell shape transformation (Figure 3E). Since depletion of Ca^2+^ from the ER usually leads to ER stress [35], we tested whether inhibition of Ca^2+^ depletion would prevent ER stress. T24 cells were pre-treated with DRN for 1 h before administering nerolidol (60 mg/L for 24 h), and GRP78 levels were determined by immunoblotting. DRN-pre-treated cells exhibited a reduction in baseline GRP78 levels and no increase after nerolidol treatment, whereas cells without DRN pre-treatment showed a strong increase in GRP78 levels (Figure 3F).

To determine the impact of high cytoplasmic Ca^2+^ levels on the mitochondria, ROS and ATP levels were measured. ROS levels were measured by the ROS-Glo H_2_O_2_ assay in TCCSUP cells 24 h after treatment with 50 mg/L nerolidol. The results showed a trend of increased ROS levels (two-tailed unpaired *t*-test, *p* = 0.0715) in nerolidol-treated cells when compared to control cells (Appendix A). ATP levels in TCCSUP cells were measured using a luciferase assay after treatment with increasing concentrations of nerolidol. A concentration-dependent reduction in ATP production was observed 20 min after treatment with nerolidol (Appendix A). After 60 min, ATP production recovered and increased in the nerolidol-treated cells in a concentration-dependent manner. This suggests that the mitochondria were not damaged and ATP production was conserved.

### 3.4. The Upstream Signals of Both Cell Death Pathways Include cAMP-Dependent Signal Transduction through β-Adrenergic Receptors, PKA, and Soluble Adenylyl Cyclases

The cAMP level was determined in TCCSUP cells treated with 50 mg/L nerolidol for 60 min using a commercial kit. The results showed a trend of increase in cAMP levels (two-tailed unpaired t-test, *p* = 0.0849) (Appendix A). Furthermore, we investigated possible sources of cAMP and its involvement in nerolidol-induced cell death. T24 cells were pre-treated with H89 (PKA inhibitor), L755507 (β-adrenergic receptor inhibitor), and KH7 (soluble adenylyl cyclase (sAC) inhibitor) for 1 h before the administration of nerolidol. To induce the early cell death pathway, cells were treated with 100 mg/L nerolidol and cell viability was assessed 2 h post-treatment. The results demonstrated the successful inhibition of early cell death by all inhibitors used (Figure 4A). All inhibitors also prevented membrane blebbing and cell swelling (Figure 4B). Although KH7 inhibited the reduction in cell viability caused by nerolidol, the cells treated with nerolidol and KH7 exhibited high cytoplasmic vacuolation, which was not caused by either compound alone. The involvement of PKA, β-adrenergic receptor, and sAC in the late cell death pathway was investigated by pre-treating the cells with inhibitors and then treating them with 60 mg/L of nerolidol for 48 h. Cell viability was assessed using cell counting and trypan blue staining. The results demonstrated that the inhibitors were successful in preventing nerolidol-induced cell death (Figure 4C) even though L755507 was significantly less effective at a higher dose (600 nM) than at a lower dose (50 pM). Cell microscopy revealed that PKA inhibition completely inhibited vacuolation and changes in cell morphology, whereas L755507 and KH7 were unable to prevent changes in cell morphology or inhibit vacuolation (Figure 4D). To determine whether the involvement of PKA, β-adrenergic receptors, and sAC was upstream or downstream of the ER stress induction, T24 cells were pre-treated with H89, L755507 (50 pM), or KH7 (10 µM) for 1 h and then treated with 60 mg/L of nerolidol for 24 h. ER stress induction was examined by immunoblotting GRP78 as a ER stress marker. The results showed that H89 completely inhibited the increase in GRP78 expression, whereas L755507 and KH7 partially inhibited the increase in GRP78 (Figure 4E).

### 3.5. Nerolidol Cytotoxicity Is Dependent on MAPK Activation Downstream of ER Stress

To test the involvement of the MAP kinase pathway in nerolidol cytotoxicity, we analyzed the activation of two MAP kinases, ERK and p38. T24 and TCCSUP cells were treated with 50 and 75 mg/L nerolidol and collected after 2, 24, and 48 h of treatment. WB showed that ERK was strongly phosphorylated after nerolidol treatment and phosphorylation increased over time in both cell lines (Figure 5A). The total expression of ERK was reduced by nerolidol treatment at the 48 h time point. Interestingly, p38 expression increased in the 50 mg/L treatment, whereas it decreased in the 75 mg/L treatment. To investigate the role of p38 in nerolidol cytotoxicity, we used SB203580, a p38 MAPK inhibitor, and assessed cell viability by cell counting and trypan blue staining. The results suggested that p38 inhibition reduced nerolidol cytotoxicity (Figure 5B). Cell microscopy revealed the presence of high cytoplasmic vacuolation in cells treated with both SB203580 and nerolidol, but not in separate treatments, at the 2 h time point (Figure 5C). p38 inhibition was effective in preventing late cell death events (Figure 5D). Cell microscopy revealed the absence of cytoplasmic vacuolation in cells pre-treated with the p38 inhibitor compared with nerolidol treatment alone (Figure 5E). Cells also proliferated slower than the control cells at a total cell count of 60 ± 5% and 85 ± 4% of the control cell counts, showing that nerolidol was still effective in reducing cell proliferation after p38 inhibition. To investigate the effect of p38 inhibition on the induction of ER stress, we pre-treated T24 cells with SB203580 (4 µM) and then treated them with 60 mg/L nerolidol for 24 h. We observed that inhibition of p38 reduced, but did not prevent, the increase in GRP78 levels (Figure 5F).

### 3.6. Short Exposure to Nerolidol Induces Similar Cytotoxic Effect on Cells as Continuous Treatments

We demonstrated that continuous exposure to nerolidol reduced cell proliferation and induced two distinct cell death events. To test whether nerolidol would produce a similar effect in a very short treatment period, we treated T24 cells with 100 mg/L nerolidol for 15, 30, or 60 min. The medium containing nerolidol was removed, cells were washed with PBS, and a fresh medium was added. Cells were counted at 1, 24, 48, and 72 h after treatment and cell viability was assessed by trypan blue staining. The results showed that short nerolidol treatment had the same overall effect on cells, although less severe than the continuous treatment (Figure 6A,B). Treating cells for 60 min, followed by 60 min of recovery, produced the same effects as 2 h of continuous treatment, inducing a high proportion of early induced cell death events, continuous loss of the total cell count, and high trypan blue staining. A 15 min treatment followed by recovery did not induce the early cell death, similar to treatment with 25 mg/L continuous treatment, but did induced late cell death. Treatment for 30 min followed by recovery induced both cell death events, with the late cell death being more pronounced. Both 15 and 30 min treatments followed by recovery periods reduced cell proliferation compared to the control cells. Morphologically, short treatments produced the same phenotype on cells as the continuous treatments (Appendix A).

### 3.7. The Two Pathways of Cell Death Differ by Their Involvement of Caspases

We observed that a fraction of nerolidol-treated TCCSUP cells stained positive for annexin V at 48 h (Figure 7A). Our aim was to evaluate the difference between two cell death pathways induced by nerolidol treatment. To test whether apoptosis is the pathway of cell death following nerolidol treatment, WB was performed at 2, 24, and 48 h post-treatment and cell death markers were analyzed. Antibodies against cleaved-caspase 3, caspase 9, and PARP1 were used as markers of apoptosis. All analyzed markers were negative (Figure 7B) as no cleaved-caspase 3, cleaved-caspase 9, or cleaved-PARP1 (~86 kDa) were found post-treatment in both T24 and TCCSUP cell lines. Interestingly, PARP1 expression was increased at 48 h in both cell lines in the 50 mg/L treatment, whereas it was decreased in the 75 mg/L treatment. Furthermore, we tested the presence of phosphorylated MLKL, a marker of necroptosis. The results showed no phosphorylation of MLKL at any of the investigated concentrations or time points. For further validation, T24 cells were incubated with z-Vad (a pan-caspase inhibitor) for 1 h before nerolidol was added and incubated for 2 or 48 h. Cell viability was assessed using cell counting and trypan blue staining. The results suggested that z-Vad was capable of preventing the early induced cell death, as shown in Figure 7C. Increasing the dose of z-Vad was more efficient and completely prevented the early induced cell death. Cell blebbing and swelling were prevented by the administration of a higher dose of z-Vad (Figure 7D). To assess the involvement of caspases in the late cell death mechanism, T24 cells were pre-treated with 40 μM z-Vad for 1 h, a concentration that completely inhibited the early induced cell death, after which cells were treated with 60 mg/L nerolidol for 48 h. Z-Vad failed to inhibit the late cell death (Figure 7E). It was also unable to prevent cytoplasmic vacuolation and cell shape transformation at multiple time points (Figure 7F). To assess the activation of caspases upstream or downstream of ER stress, T24 cells were pre-treated with 40 μM z-Vad for 1 h and then treated with 60 mg/L of nerolidol for 2 h. GRP78 protein expression was increased in both uninhibited and z-Vad-treated cells, which demonstrated the involvement of caspases downstream of ER stress (Figure 7G).

## 4. Discussion

In this study, we explored the signaling cascade involved in nerolidol cytotoxicity in bladder carcinoma cell lines. We demonstrated that nerolidol induces the displacement of Ca^2+^ from the ER through RYR Ca^2+^ channels, thus inducing ER stress and confirming the central role of Ca^2+^ in nerolidol cytotoxicity. RYR Ca^2+^ channels are usually expressed in muscle cells and cells of the central nervous system, but they are also expressed in urothelial cells [36,37]. RYR Ca^2+^ channels can be activated through multiple mechanisms [38], including phosphorylation by PKA [38], a cAMP-dependent protein kinase. We found that PKA inhibition by the H89 inhibitor completely prevented nerolidol cytotoxicity and other phenotypic changes. PKA is activated by cAMP synthesized by adenylyl cyclizes (AC), which are typically located on the cell membrane and can be activated by a large number of membrane receptors [39,40,41]. Inhibition of β-adrenergic receptors was capable of inhibiting nerolidol cytotoxicity, but was not able to completely inhibit changes in cell morphology. This finding implies that in addition to β-adrenergic receptors, there must be another inducer of cAMP production located upstream of PKA. We identified it in the form of soluble AC (sAC), a member of the AC family that, unlike membrane-bound AC, is not activated by G-proteins but rather by an increase in Ca^2+^ and bicarbonate ions in the cytoplasm [42]. Inhibition of sAC prevented nerolidol cytotoxicity; however, similar to the inhibition of β-adrenergic receptors, it was not able to completely inhibit changes in cell morphology. Considering these results, it appears that nerolidol activates β-adrenergic receptors, which triggers cAMP production, thus activating PKA. PKA phosphorylates RYR and Ca^2+^ is released from the ER into the cytoplasm. Increased cytoplasmic levels of Ca^2+^ activate sAC, which additionally produces cAMP, potentiates PKA activation, and sustains Ca^2+^ release from the ER. We further demonstrated that downstream signal transduction is mediated by two MAP kinases, ERK and p38. ERK was strongly and increasingly phosphorylated after nerolidol treatment, which persisted for the duration of the experiment. Inhibition of p38 prevented nerolidol cytotoxicity and induced cytoplasmic vacuolation, which is indicative of the activation of autophagy [43]. Autophagy was activated between the two cell death periods, as shown by an increase in LC3 protein expression and processing at 24 h after nerolidol treatment. At this time point, cell death was less prevalent, implying that the activation of autophagy has a pro-survival function [44]. Interestingly, inhibition of the described signaling cascade inhibited both cell death pathways, suggesting that they share a dependence on Ca^2+^ release from the ER and activation of ER stress. 

Nerolidol administration induced two distinct cell death pathways with distinct changes in cell morphology and a shared initial signaling cascade. Early induced cell death occurred shortly after exposure to nerolidol and was characterized by cell blebbing and swelling, which led to cell membrane rupture. Late-induced cell death started at 48 h post-treatment and was preceded by cell shape transformation and cytoplasmic vacuolation. The observed cytoplasmic vacuolation is indicative of ER stress, which can lead to cell death [35,45,46,47,48]. ER stress can be induced by a myriad of factors, including Ca^2+^ imbalance and the accumulation of unfolded proteins [47,49], where the latter was previously attributed to nerolidol [19]. Apart from the observed morphological changes in cells, these two types of cell death differed in the involvement of caspases. We and others [20] found no activation of the caspase cascade involved in canonical apoptosis [50] following nerolidol treatment. In contrast, the pan-caspase inhibitor, z-Vad, successfully inhibited the early induced cell death, demonstrating the involvement of unidentified caspase/s. One of the cell death pathways that fits perfectly with the described signaling cascade, specific morphological features (cell blebbing, swelling, and membrane rupture), and involvement of caspases is pyroptosis (Appendix A). Pyroptosis is a caspase-dependent necrotic cell death pathway [51,52]. It requires the creation of an inflammasome [51,52], which leads to the cleavage of caspase 1, explaining the effectiveness of z-Vad [52]. Ca^2+^ has been identified as the main contributor to membrane blebbing and cell swelling during pyroptosis [52]. p38 has also been implicated in pyroptosis and inflammasome activation [53]. On the other hand, the late cell death mechanism resembles paraptosis. Paraptosis is a caspase-independent cell death pathway [54], characterized by cytoplasmic vacuolation [47], and is triggered by the depletion of Ca^2+^ from the ER to the cytoplasm, causing ER stress [45,55]. Paraptosis is mediated by ERK and its prolonged activation [56] owing to increased ROS production [57]. Similar to apoptosis, paraptosis is annexin positive, but it does not stain with PI, as cells retain intact cell membranes [19]. This is usually due to prolonged high Ca^2+^ levels in the cytoplasm that activate scramblases [58]. Biazi et al. [19] described the expression profile characteristics of paraptosis-mediated cell death after nerolidol treatment, which complements our findings (Appendix A). It is unclear why a fraction of cultured cells treated with nerolidol undergo early induced cell death, while others survive until the late cell death period. This could be due to the position of each cell in the cell cycle, which creates different sensitivity and susceptibility to nerolidol cytotoxicity. Tumor cells actively proliferate and are distributed in different stages of the cell cycle. Understanding the mechanisms of two distinct cell deaths induced by nerolidol will increase the benefit of nerolidol as a potential anti-cancer chemotherapeutic. In addition to inducing cell death, nerolidol reduced cell proliferation, which is another beneficial effect that can be exploited for treatment. The reduced expression of cell cycle markers, as well as the presence of DNA damage, could explain the reduced cell proliferation [59], which is in accordance with previous findings [19,60]. 

ER stress has been observed in bladder cancer as a form of tumor adaptation to its microenvironment and applied treatment, leading to the creation of resistance and more aggressive behavior [32]. In contrast, high ER stress leads to cell death [32,35]. β-adrenergic receptors are expressed in urothelial cells [61,62] and overexpressed in tumors [63,64]. Tumor cells utilize this signaling cascade to increase cell proliferation, induce epithelial-to-mesenchymal transition and angiogenesis, evade the immune response, and invade tissues [64]. The use of antagonists of β-adrenergic receptors is currently being tested against tumors [63]. The ability of nerolidol to overstimulate β-adrenergic receptors and cause cell death provides an opportunity. Killing tumor cells through the mechanism they use for their survival [64]. Nerolidol was successful in combating BC at the cell culture level; however, further investigation is needed to prove its true potential. Through its unorthodox induction of cell death, which is independent of the apoptotic mechanism, it could be used to combat resistant BC cells that cause BC recurrence and disease progression. The biggest obstacle to the utilization of nerolidol in systemic chemotherapy is its low solubility and high affinity for incorporation into cell membranes [2,24], which would deplete nerolidol before it reaches the tumor cells. In our opinion, the best way to utilize nerolidol would be to treat the tumor site directly, either by injecting the tumor site or administering therapy in the arterial system that feeds the tumor. BC provides a rare opportunity for which surgical application would not be needed. Nerolidol could be administered intravesically, similar to classical chemotherapy [31], where its high affinity for incorporation into the cell membrane would ensure that it reaches deep into the tumor-affected area. This mode of application could be best utilized against NMIBC, which is completely located in the bladder and thus can be immersed in nerolidol treatment. We demonstrated that the nerolidol-activated signaling cascade that leads to cytotoxicity becomes independent of nerolidol treatment after a short treatment period (15–60 min), confirming the creation of a feedback loop, where the release of Ca^2+^ from the ER to the cytoplasm activates sAC, which potentiates PKA activity, further sustaining the release of Ca^2+^ from the ER and continuing the signal. Brief treatment (15–60 min) with nerolidol reduced cell proliferation and induced cell death, which is in accordance with the usual length of intravesical treatment (30–120 min) [30]. Although our study was based on cell culture, we demonstrated that nerolidol could be a promising candidate for future investigation as a BC therapy.

## 5. Conclusions

The cytotoxic effect of nerolidol results in the activation of two distinct cell death pathways in bladder cancer cell lines. The cytotoxicity is mediated through β-adrenergic receptor signaling, activation of PKA, and soluble AC, which led to the secretion of Ca^2+^ from the ER into the cytoplasm through RYR channels, resulting in ER stress. DNA damage accumulates as a consequence of the ER stress response and induces cell cycle arrest. The findings presented in this study highlight nerolidol as a promising candidate for bladder cancer therapy.

## Figures and Tables

**Figure 1 cancers-15-00981-f001:**
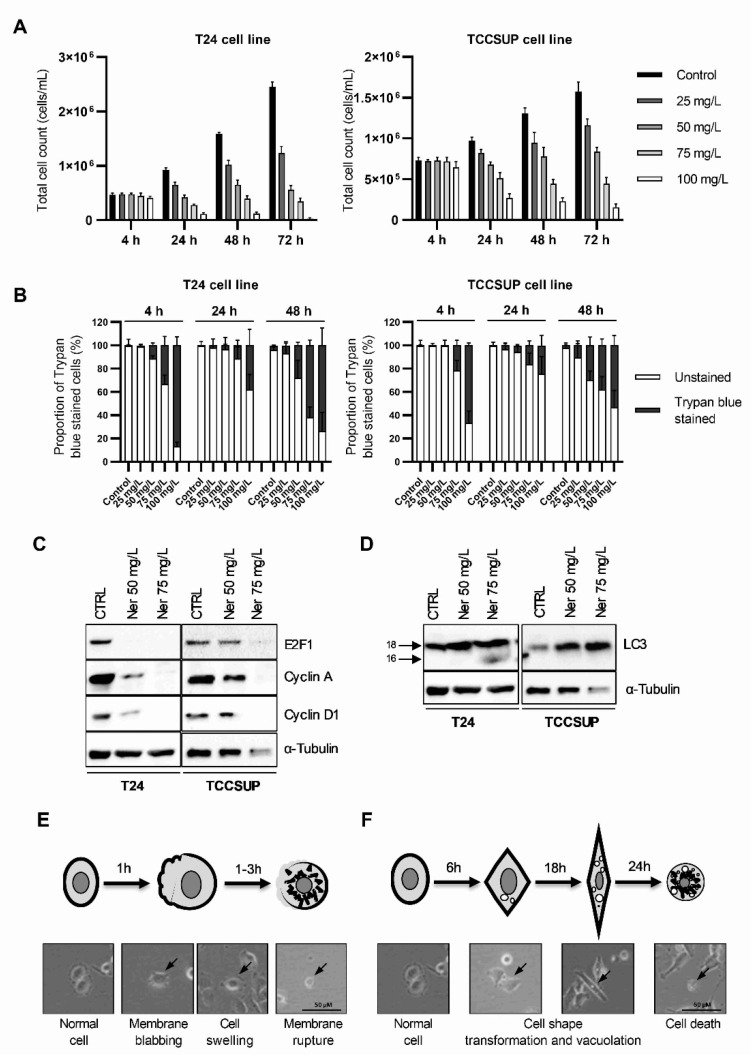
Nerolidol induces two cell death periods with distinct cell morphologies. (**A**) T24 (left) and TCCSUP (right) total cell counts at 4, 24, 48, and 72 h after nerolidol was administered at 25, 50, 75, and 100 mg/L. (**B**) T24 (left) and TCCSUP (right) cell viability was determined at 4, 24, and 48 h after nerolidol was administered at 25, 50, 75, and 100 mg/L. Cell viability was determined by cell counting after trypan blue staining and presented as a share of stained cells in the total cell count (100%) for each measurement. Stained cells were considered not viable. (**C**) Immunostaining of cell cycle markers E2F1, cyclin A, and cyclin D1 in T24 and TCCSUP cell lines after treatment with 50 and 75 mg/L of nerolidol for 24 h analyzed by WB. (**D**) Expression and processing of autophagy marker LC3 in T24 and TCCSUP cell lines after treatment with 50 and 75 mg/L of nerolidol for 24 h analyzed by WB. (**E**) Schematic representation of changes in cell morphology that describe early induced cell death with an indicated timeline corroborated by cell photographs. (**F**) Schematic representation of changes in cell morphology that describe late cell death event with an indicated timeline corroborated by cell photographs (arrows indicate cells with described morphological features). The uncropped blots and molecular weight markers are shown in Appendix A.

**Figure 2 cancers-15-00981-f002:**
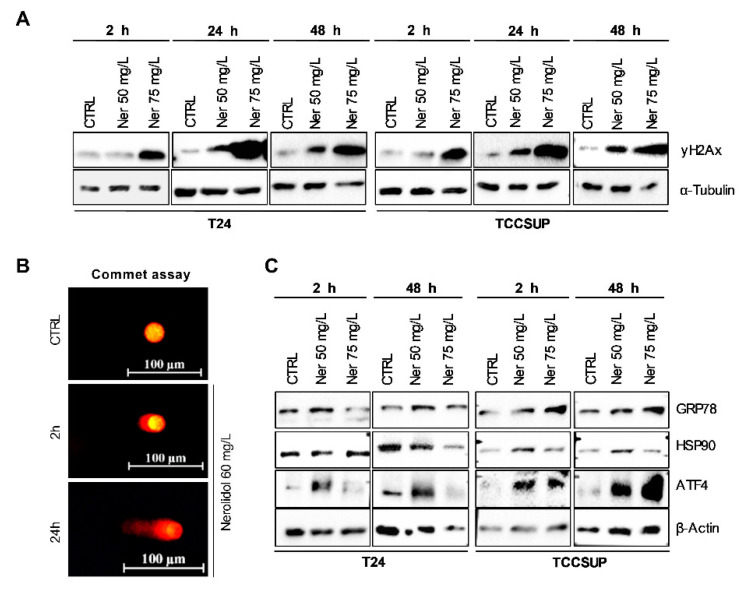
Prolonged nerolidol exposure caused increasing levels of DNA damage and ER stress. (**A**) γH2Ax immunostaining on T24 (left) and TCCSUP (right) cells after treatment with 50 and 75 mg/L of nerolidol for 2, 24, and 48 h. (**B**) Representative images of comet assay results on T24 cells treated with 60 mg/L of nerolidol for 2 and 24 h. (**C**) Immunostaining of ER stress markers GRP78, HSP90, ATF4, and GADD153 (CHOP) on T24 (left) and TCCSUP (right) cells after treatment with 50 and 75 mg/L of nerolidol for 2 and 48 h. The uncropped blots and molecular weight markers are shown in Appendix A.

**Figure 3 cancers-15-00981-f003:**
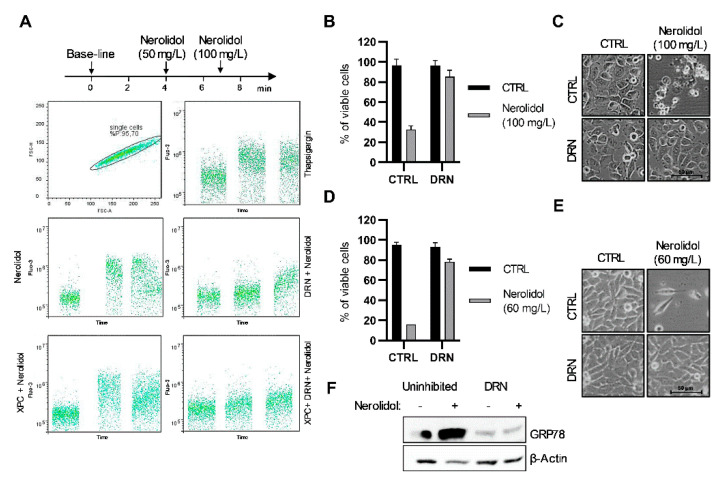
Nerolidol cytotoxicity originates from Ca^2+^ depletion in the ER through ryanodine receptor (RYR). (**A**) Determination of cytoplasmic Ca^2+^ level by labelling with the Fluo-3 Ca^2+^ stain and measured by flow cytometry. Single-cell suspension of TCCSUP cells was prepared in Ringer solution (containing Ca^2+^) for thapsigargin treatment and PBS (without Ca^2+^) for nerolidol treatments. Cells in PBS were pre-treated for 1 h with XSC (380 nM), DRN (10 μM), or their combination. Base-line Ca^2+^ level was determined for each combination followed by administration of 50 mg/L of nerolidol. An additional dose of 100 mg/L of nerolidol was administered in order to provoke a maximal increase in Ca^2+^ level in the cytoplasm. Thapsigargin was used as a positive control. (**B**) Cell viability of T24 cells pre-treated with DRN (10 μM) for 1 h and then treated with 100 mg/L of nerolidol for 2 h determined by cell counting and trypan blue staining. Cell counts were normalized to control cells. (**C**) Cell microscopy images of T24 cells pre-treated with DRN (10 μM) for 1 h and then treated with 100 mg/L of nerolidol for 2 h. (**D**) Cell viability of T24 cells pre-treated with DRN (10 μM) for 1 h and then treated with 60 mg/L of nerolidol for 48 h determined by cell counting and trypan blue staining. Cell counts were normalized to control cells. (**E**) Cell microscopy images of T24 cells pre-treated with DRN (10 μM) for 1 h and then treated with 60 mg/L of nerolidol for 48 h. (**F**) T24 cells treated with DRN (10 μM) for 1 h before nerolidol treatment (60 mg/L 24 h). Immunostaining of GRP78 was used to investigate the inhibitory effect on ER stress induction. The uncropped blots and molecular weight markers are shown in Appendix A.

**Figure 4 cancers-15-00981-f004:**
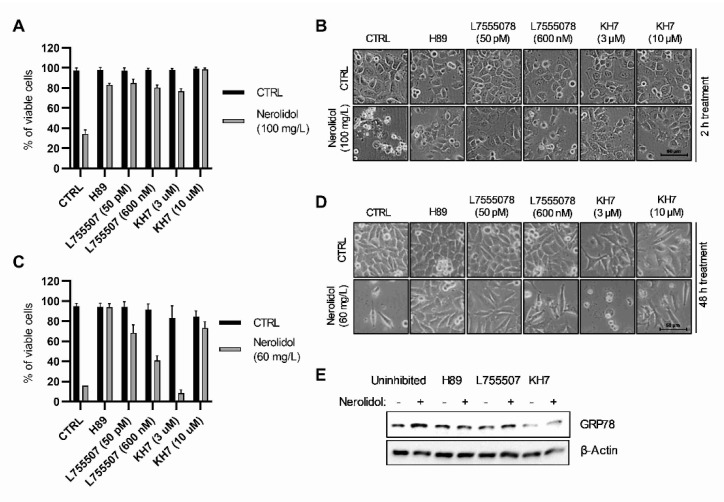
Both nerolidol-induced cell death pathways include cAMP-dependent signal transduction through β-adrenergic receptors, PKA, and soluble adenylyl cyclases. (**A**) Cell viability of T24 cells pre-treated with H89 (50 nM), L755507 (50 pM and 600 nM), and KH7 (3 and 10 μM) for 1 h and then treated with 100 mg/L of nerolidol for 2 h determined by cell counting and trypan blue staining. Cell counts were normalized to control cells. (**B**) Cell microscopy images of T24 cells pre-treated with H89 (50 nM), L755507 (50 pM and 600 nM), and KH7 (3 and 10 μM) for 1 h and then treated with 100 mg/L of nerolidol for 2 h. (**C**) Cell viability of T24 cells pre-treated with H89 (50 nM), L755507 (50 pM and 600 nM), and KH7 (3 and 10 μM) for 1 h and then treated with 60 mg/L nerolidol for 48 h determined by cell counting and trypan blue staining. Cell counts were normalized to control cells. (**D**) Cell microscopy images of T24 cells pre-treated with H89 (50 nM), L755507 (50 pM and 600 nM), and KH7 (3 and 10 μM) for 1 h and then treated with 60 mg/L of nerolidol for 48 h. (**E**) T24 cells treated with H89 (50 nM), L755507 (50 pM), and KH7 (10 μM) for 1 h before administration of 60 mg/L nerolidol for 24 h. Immunostaining of GRP78 was used as a marker of ER stress activation. The uncropped blots and molecular weight markers are shown in Appendix A.

**Figure 5 cancers-15-00981-f005:**
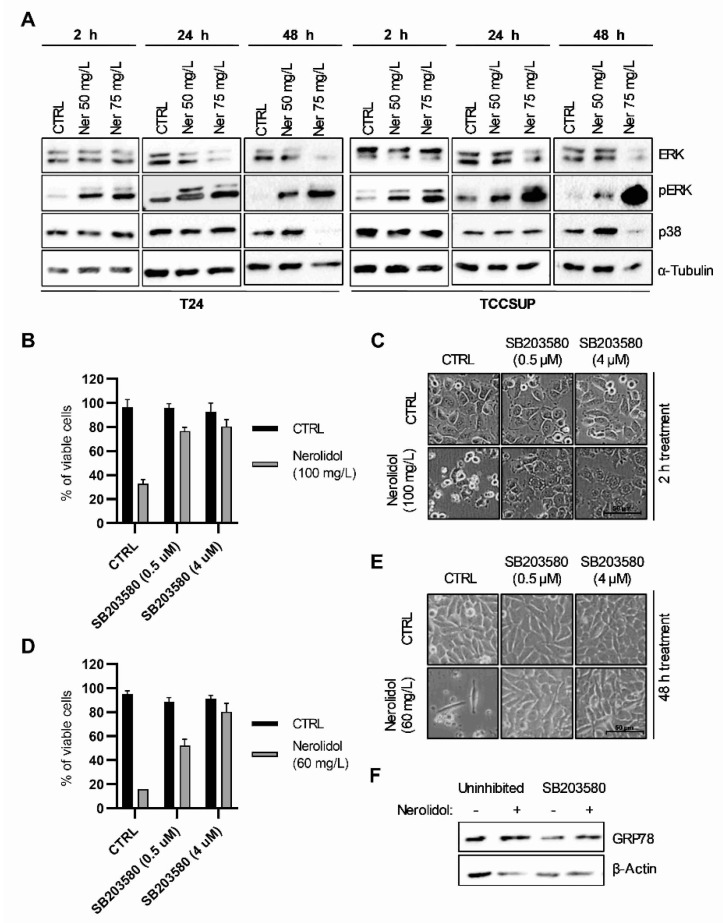
Nerolidol cytotoxicity is driven through MAPKs. (**A**) Immunostaining of ERK, pERK, and p38 on T24 (left) and TCCSUP (right) cells after treatment with 50 and 75 mg/L of nerolidol for 2, 24, and 48 h. α-tubulin was used as a loading control. (**B**) Cell viability of T24 cells pre-treated with SB203580 (0.5 and 4 μM) for 1 h and then treated with 100 mg/L of nerolidol for 2 h determined by cell counting and trypan blue staining. Cell counts were normalized to control cells. (**C**) Cell microscopy images of T24 cells pre-treated with SB203580 (0.5 and 4 μM) for 1 h and then treated with 100 mg/L of nerolidol for 2 h. (**D**) Cell viability of T24 cells pre-treated with SB203580 (0.5 and 4 μM) for 1 h and then treated with 60 mg/L of nerolidol for 48 h determined by cell counting and trypan blue staining. Cell counts were normalized to control cells. (**E**) Cell microscopy images of T24 cells were pre-treated with SB203580 (0.5 and 4 μM) for 1 h and then treated with 60 mg/L of nerolidol for 48 h. (**F**) T24 cells treated with SB203580 (4 μM) for 1 h before nerolidol (60 mg/L) administration and incubated for 24 h. Immunostaining of GRP78 was used to test the effect on ER stress induction. The uncropped blots and molecular weight markers are shown in Appendix A.

**Figure 6 cancers-15-00981-f006:**
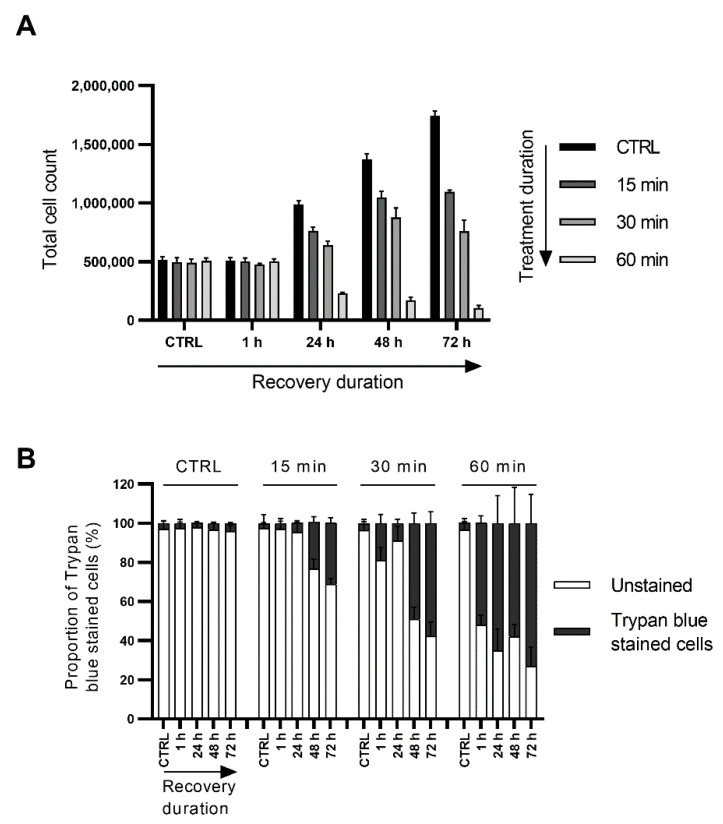
Short exposure to nerolidol reduces cell proliferation and induces both cell death events. (**A**) Total cell count from T24 cells after a short treatment with 100 mg/L of nerolidol (15, 30, or 60 min) followed by a recovery period of 1, 24, 48, and 72 h. (**B**) Cell viability of T24 cells after a short treatment with 100 mg/L of nerolidol (15, 30, or 60 min) followed by a recovery period of 1, 24, 48, and 72 h. Cell viability was assessed by trypan blue staining and presented as the share of stained cells in the total cell count (100%) for each measurement. Stained cells were considered not viable.

**Figure 7 cancers-15-00981-f007:**
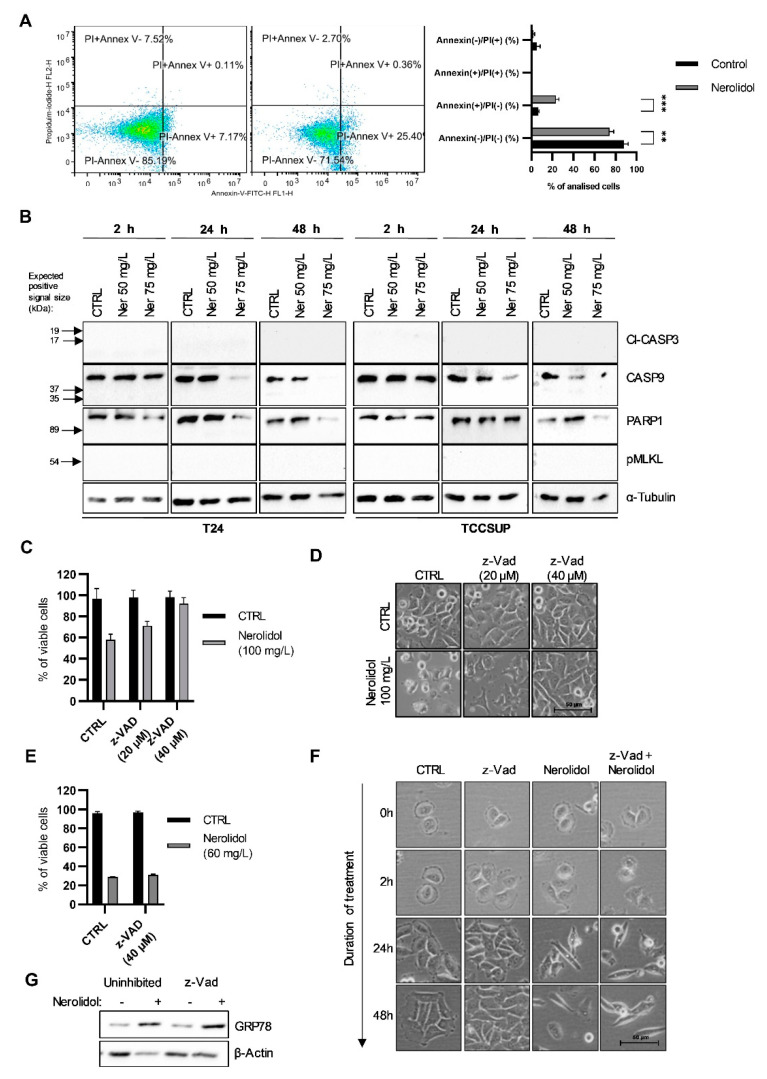
The two pathways of cell death differ by their involvement of caspases. (**A**) Flow cytometry results of annexin V/PI staining on TCCSUP cells untreated (left) or treated (right) with 50 mg/L of nerolidol for 48 h. The chart represents the quantification of two independent experiments. (**B**) Immunostaining of apoptotic and necroptotic factors with cl-CASP3, CASP9, PARP1, and pMLKL on T24 (left) and TCCSUP (right) cells after treatment with 50 and 75 mg/L of nerolidol for 2, 24 and 48 h. (**C**) Cell viability of T24 cells pre-treated with z-Vad (20 and 40 μM) for 1 h and then treated with 100 mg/L of nerolidol for 2 h determined by cell counting and trypan blue staining. Cell counts were normalized to control cells. (**D**) Cell microscopy images of T24 cells pre-treated with z-Vad (20 and 40 μM) for 1 h and then treated with 100 mg/L of nerolidol. (**E**) Cell viability of T24 cells pre-treated with z-Vad (40 μM) for 1 h and then treated with 60 mg/L of nerolidol for 48 h determined by cell counting and trypan blue staining. Cell counts were normalized to control cells. (**F**) Cell microscopy images of T24 cells pre-treated with z-Vad (40 μM) for 1 h and then treated with 60 mg/L of nerolidol taken at multiple time points following nerolidol administration. (**G**) T24 cells were treated with z-Vad (40 μM) for 1 h before 60 mg/L nerolidol was administered for 2 h. Immunostaining of GRP78 was used as a marker of ER stress induction. not significant (ns) for *p* > 0.05, **: *p* < 0.001, ***: *p* < 0.0001. The uncropped blots and molecular weight markers are shown in Appendix A.

## Data Availability

All data are available in the main text or the Appendix A.

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
