# Peer review of "Mechanism of cis-Nerolidol-Induced Bladder Carcinoma Cell Death"

_cancers, 2023, doi:10.3390/cancers15030981_

Round 1
Reviewer 1 Report
The topic is of interest, and the manuscript is well illustrated.
Major Comments:
1. Are there controversies in this field? What are the most recent and important achievements in the field? In my opinion, answers to these questions should be emphasized. Perhaps, in some cases, novelty of the recent achievements should be highlighted by indicating the year of publication in the text of the manuscript.
2. The results and discussion section is very weak and no emphasis is given on the discussion of the results like why certain effects are coming in to existence and what could be the possible reason behind them?
3. Conclusion: not properly written.
4. Results and conclusion: The section devoted to the explanation of the results suffers from the same problems revealed so far. Your storyline in the results section (and conclusion) is hard to follow. Moreover, the conclusions reached are really far from what one can infer from the empirical results.
5. The discussion should be rather organized around arguments avoiding simply describing details without providing much meaning. A real discussion should also link the findings of the study to theory and/or literature.
6. Spacing, punctuation marks, grammar, and spelling errors should be reviewed thoroughly. I found so many typos throughout the manuscript.
7. English is modest. Therefore, the authors need to improve their writing style. In addition, the whole manuscript needs to be checked by native English speakers.
Reviewer 2 Report
I am writing about the paper entitled 'Mechanism of cis-nerolidol-induced bladder carcinoma cell death'. The scientific idea behind this paper is good. But I have major concerns related to this paper.
There are so many literature available related to the activity of Nerolidol in invite and in vitro. So what is the nobel factor of this paper? Author used only 2 cell lines in the whole paper. Author should used an In vivo model to justify their findings. The methods author used in this paper is not sufficient to justify this findings.
For all the microscopic experiments, author should add the images in the main manuscript.
1. in figure 1A and B, author used 25-100mg.l nerolidol. But in the method section they mentioned 25-200mg/ml. in the figure 200mg/ml is missing.
From the figure 1B it is clearly sen that 100mg/ml concentration had higher number of cell death. but in the whole manuscript author mostly used 50mg/ml conc. How did author chose this concentration? did author tested IC50 of this compound? should include.
For figure 1B, its unclear in the method section, how author counted the cells and make this figure.
2.Trypan blue staining images are needed to add in the manuscript for different conc. with different time points.
3. Author should provide all the RAW western blots to justify the findings.
4. Schematic representation of the figure 1E and 1F is not clear. Need to add some writings or words to indicate the events.
5. Through out the manuscript, the conc. of Nerolidol is not consistent. Why?
Reviewer 3 Report
This research Mechanism of Cis-Nerolidol-Induced Bladder Carcinoma Cell 2 Death by
Glumac et. al. investigated the antitumor activity and mechanism of nerolidol in bladder cancer cell line model. The manuscript is well-organized and clearly written and the results are thoroughly discussed. However, in the current study, the author used different treatment doses of nerolidol (50 mg/l, 60 mg/l, 75 mg/l) in different experiments, without mentioning the significance of the experiment-selected dose concentration used. Second, there is an issue with the result analysis as compared to the Immunoblot figures.
1. Line 151: There is a typo error. There is extra space at the start of the last line.
2. Line 208: TSS SUP cells need to be replaced as used in the entire manuscript TCCSUP cells.
3. Figure 2A: Why there is a change in the gamma H2AX level in the CTRL group in 2h vs 24 h vs 48 h in both the cell lines. There is no treatment in the CTRL group and the figure showed a change in the gamma H2AX level which indicates stressed cells.
4. Figure 2A: Why there is a decrease in the gamma H2AX in 48h treatment (TCCSUP cells) as compared to the CTRL group; but increased in T24 cells.
5. Figure 2A: Loading control is not equal, interfering with the result analysis. It is suggested that the improved western blot result will be done and a reanalysis of the gamma H2AX level will be determined.
6. Why Nerolidol (60 mg/l) was used in comet assay instead of 50mg/l or 75mg/l.
7. Figure 2C: Immunoblot image is not clean; it is difficult to analyze the result from this figure. It is suggested that this experiment must be repeated and reanalyzed.
8. Line 322: sup. Figure 2 C is not reflecting the result statement.
9. Line 351: Needs to remove extra space: death. T24 cells were…
10. Line 371-373: result statement for Figure 4E is not reflecting the correct analysis of the result. There is an increase in the expression of GRP78 instead of a decrease as mentioned by the author.
11. Figure 4 E: Loading control shows the differential expression.
12. Line 394: The total expression of ERK and p38 was not significantly affected by 394 nerolidol treatment. However, comparing the figure result shows differential expression of ERK and p38 protein levels.
13. Figure 5A: Loading control is not equal. What dual ERK band indicates? Why there is an increase in pERK level in the 24h CTRL group in both the cell lines?
14. Line 422: Result heading: 3.4 and 3.5 are the same: Nerolidol cytotoxicity is driven through MAPKs.
15. Result: 3.5 Nerolidol cytotoxicity is driven through MAPKs, there is no result in the support of the heading title.
16. Figure 7B: As there is no cleavage observed, why there is a decrease in the expression of CAS9 and PARP1 in both the cell lines? Also loading control is not equal.
17. Line 556: [54]. Similar…. needs to replace with [54]. Similar…..
18. Line 558: scramblases [55]. needs to replace with scramblases [55].
19. References: Many of the references (5, 6, 7, 14, 18, 20, 22, 39, and 51) missing the page number.
20. Line 623: Reference 7 needs to be correct according to the Cancers Journal format.
Round 2
Reviewer 1 Report
1. The discussion should be rather organized around arguments avoiding simply describing details without providing much meaning. A real discussion should also link the findings of the study to theory and/or literature.
2. English is modest. Therefore, the authors need to improve their writing style. In addition, the whole manuscript needs to be checked by native English speakers.
Author Response
1. We appreciate the reviewer's comments and concerns related to the concept of the Discussion section. Although describing findings from the manuscript in the Discussion section is not always preferred (or even needed), we argue that in this case, it is necessary, in order to emphasize what exactly was found in the study and what was the purpose of each experiment. Each of the studied signaling factors had its purpose in elucidating the final signaling cascade and a type of activated cell death mechanism. The primary focus of our study was to shed light on the mechanism of nerolidol signaling and link this signal to cellular cytotoxicity.
2. We appreciate the reviewer's comment. We have re-checked the manuscript for errors and have made improvements to the writing style. We believe that now the manuscript is easy to read and understand.
Reviewer 2 Report
Dear author thanks for addressing my concerns. All the questions are addressed very well except I didn't see any RAW file of western blots.
Round 3
Reviewer 2 Report
Dear Author,
Thank you so much for addressing my concerns.
